# EEG-Based Neurofeedback in Athletes and Non-Athletes: A Scoping Review of Outcomes and Methodologies

**DOI:** 10.3390/bioengineering12111202

**Published:** 2025-11-03

**Authors:** Rui Manuel Guerreiro Zacarias, Darshika Thejani Bulathwatta, Ilona Bidzan-Bluma, Saúl Neves de Jesus, João Mendonça Correia

**Affiliations:** 1 University Research Center in Psychology (CUIP), Faculty of Human and Social Sciences (FCHS), University of Algarve, 8005-139 Faro, Portugal; snjesus@ualg.pt (S.N.d.J.); jmcorreia@ualg.pt (J.M.C.); 2 Institute of Psychology, University of Gdańsk, 80-309 Gdańsk, Poland; bdibul@ou.ac.lk (D.T.B.); ilona.bidzan-bluma@ug.edu.pl (I.B.-B.)

**Keywords:** neurofeedback, EEG, sham control, reproducibility, cognitive performance, sports performance

## Abstract

**Background**: Electroencephalography (EEG) is a non-invasive technique that records millisecond-scale cortical electrical activity using scalp electrodes. In EEG-based neurofeedback (NFB), these signals are processed to provide real-time feedback that supports self-regulation of targeted brain rhythms; evidence suggests improvements in cognitive and neurophysiological performance in athletes and non-athletes. However, methodological inconsistencies—such as limited blinding, poor sham control, and outdated approaches to EEG spectral analysis—restrict reproducibility and hinder cumulative progress in the field. **Methods**: This scoping review aimed to identify and analyze the methodological characteristics, outcome measures, and reproducibility gaps in EEG-based NFB studies involving athletes and non-athletes. Following PRISMA-ScR guidelines, we systematically searched academic databases (PubMed, Embase, Scopus, Web of Science, PsycINFO, and Cochrane Library), as well as gray literature sources (ProQuest Dissertations, LILACS, Tripdatabase, and Google Scholar). Of 48 included studies, 44 were published in international peer-reviewed journals and 4 in regional journals. Data were extracted on study design, participant population, NFB protocols, targeted EEG rhythms, cognitive and neurophysiological outcomes, and methodological rigor. **Results**: The review revealed substantial heterogeneity in targeted rhythms, protocols, and reporting standards. None of the studies employed modern spectral parameterization methods (e.g., FOOOF), while only 29% used active sham protocols and 6% employed inert sham conditions. Reporting blinding procedures and follow-up assessments was limited or absent in most studies. **Discussion**: This review highlights critical methodological shortcomings that may bias interpretations of NFB effects in sport and cognitive domains. To strengthen future research, studies should rigorously implement sham and blinding procedures, ensure transparent reporting of EEG metrics, and adopt open-science practices, including modern approaches to spectral parameterization.

## 1. Introduction

Electroencephalogram-based neurofeedback (EEG-NFB) has emerged as a promising non-invasive intervention to enhance cognitive and psychophysiological functioning, including attention, emotion regulation, and motor preparation [1,2,3,4]. In sports, NFB is increasingly applied to improve performance under pressure and support resilience across disciplines such as football, archery, judo, and swimming [5].

Despite encouraging findings, such as improvements in attention, emotion regulation, and athletic performance reported in previous EEG-NFB studies [2,3,4,5], the current evidence is constrained by major methodological limitations. In the sports context, EEG-NFB has been increasingly applied to enhance attentional focus, optimize sensorimotor rhythm regulation, and support stress management during competition. Studies have reported improvements in accuracy (e.g., archery, shooting), faster reaction times, and decision-making in football, highlighting its potential to strengthen both cognitive and motor domains in athletes [6,7,8,9,10,11,12,13,14,15,16,17].

Studies differ in protocol duration, targeted brain regions, and outcome measures, while most lack rigorous sham or double-blind designs, raising concerns about expectancy and placebo effects [18,19]. Such inconsistencies limit the attribution of NFB-induced changes to genuine neurophysiological mechanisms.

To address these challenges, the CRED-nf checklist [20] established standards for study design and reporting, including preregistration, detailed feedback specifications, sham controls (active and inert), and transparent reporting of outcomes. However, adherence remains inconsistent, and many studies still rely on closed-source analysis pipelines. Proprietary implementations of Fast Fourier Transform (FFT) parameters—such as window length or artifact rejection—are rarely disclosed, undermining reproducibility [21,22].

Notably, none of the reviewed studies employed modern spectral parameterization approaches, such as Fitting Oscillations & One-Over-F (FOOOF [23]), which separate periodic and aperiodic components to strengthen neurophysiological validity. This methodological gap is especially critical for sports applications, where subtle cognitive and performance-related changes demand precise measurement [2,24].

Finally, the drive for ecological validity—defined as the extent to which experimental findings can be generalized to real-world contexts—has led to portable EEG systems and semi-natural group protocols [13,25,26,27]. These approaches aim to capture cognitive and neurophysiological processes in more authentic environments, such as practice and competition, thereby increasing the applicability of research findings. While promising, they also introduce procedural challenges, including greater susceptibility to noise and artifacts, which require careful methodological control.

The aim of this scoping review is to systematically map methodological and analytical gaps in EEG-NFB studies, evaluate the current state of interventions in both athletes and non-athletes, and identify priorities for advancing transparency, reproducibility, and neurophysiological validity in future research.

## 2. Materials and Methods

### 2.1. Protocol and Registration

This scoping review was conducted in accordance with the Preferred Reporting Items for Systematic Reviews and Meta-Analyses extension for Scoping Reviews (PRISMA-ScR [28]). The protocol was developed a priori, following PRISMA-P guidelines [29] and methodological recommendations from the Joanna Briggs Institute [30].

To ensure transparency, reproducibility, and methodological rigor, the review was prospectively registered on the Open Science Framework (OSF). The protocol defines the eligibility criteria, outlines procedures for study selection, data extraction, and synthesis, and specifies the use of Rayyan software for independent screening by two reviewers [31].

The complete protocol is publicly accessible under the DOI registration: https://doi.org/10.17605/OSF.IO/XCUWY (accessed on 22 October 2025).

### 2.2. Eligibility Criteria

The research question was developed using the PCC framework—Population, Concept, and Context [30]. The target population included adults ≥ 18 years, divided into three groups: (i) elite athletes, training and competing at professional or international levels, with weekly physical activity typically >9 METs (vigorous-intensity [32]; (ii) amateur athletes, engaged in regular but non-professional sports practice, typically 3–9 METs; and (iii) non-athletes, healthy adults without organized sport participation, typically <3 METs. These classifications were based on the Compendium of Physical Activities [32].

The concept focused on EEG-NFB as the primary intervention. Only studies reporting objective neurophysiological or cognitive outcomes were eligible (e.g., event-related potentials (ERP), quantitative electroencephalography (qEEG), low-resolution electromagnetic tomography (LORETA); or validated measures of attention, working memory, reaction time). Studies based solely on self-reported questionnaires or satisfaction ratings were excluded.

The context included sports and laboratory settings. Eligible studies could come from any country, year, or language, provided full-text access and accurate translation into English or Portuguese.

Only original empirical studies were included randomized controlled trials (RCTs), quasi-experimental, cohort, observational, qualitative, or mixed-method studies with explicit NFB interventions. Exclusions comprised systematic reviews, meta-analyses, theoretical papers, editorials, and conference abstracts—although these were screened for additional references.

Studies were excluded if they: (i) involved clinical populations, (ii) failed to report detailed NFB protocols or outcomes, (iii) assessed only non-specific effects (e.g., expectancy, placebo), or (iv) lacked peer-review. Full-text availability was mandatory.

### 2.3. Information Sources and Search Strategy

The research team conducted an extensive literature search across seven academic databases: PubMed/MEDLINE, Embase, Scopus, Web of Science, PsycINFO, Cochrane Library, and LILACS. Both controlled vocabulary terms (MeSH, DeCS) and free-text terms were used to link NFB with EEG, cognitive performance, ERP, qEEG, and LORETA. Boolean operators and truncations were adapted for each database.

To minimize publication bias, grey literature sources were also searched, including Google Scholar, ProQuest Dissertations & Theses, Trip Database, and Dissertations Citation Index. Reference lists of included studies were hand-searched to identify additional articles.

No restrictions were applied regarding year, language, or country of origin, provided accurate translation could be ensured. References were deduplicated in EndNote X9 (Clarivate Analytics), and records were screened independently by two reviewers using Rayyan [31].

The complete database search strategies, including the detailed list of keywords applied, are reported in the Appendix A.

### 2.4. Study Selection Process

The Rayyan system (Qatar Computing Research Institute) was employed to conduct the study selection process in two phases. In the first phase, two independent reviewers (RZ and TB) screened titles and abstracts against the eligibility criteria. In the second phase, the same reviewers conducted a full-text evaluation of studies that passed the initial screening. Disagreements were resolved through discussion, and when consensus could not be reached, a third reviewer (IBB) acted as arbitrator.

Additionally, the reference lists of included studies were manually reviewed to identify further eligible records. The entire selection process was documented through the PRISMA-ScR flow diagram, including reasons for exclusion during the full-text stage. To enhance methodological rigor, the procedure underwent independent double verification, thereby ensuring transparency, reliability, and reproducibility.

### 2.5. Data Charting Process and Data Items

The first reviewer (RZ) independently performed the data charting process using a structured extraction form based on the PCC framework. A second reviewer (TB) verified all extracted data, while the first reviewer (RZ) and a third reviewer (IBB) resolved any discrepancies through discussion.

The extraction process included study characteristics (authors, year, country, study design), population details (type of participants: elite athletes, amateur athletes, or non-athletes; age range; gender distribution; and level of competition), characteristics of the NFB intervention (protocol type, frequency band, number and duration of sessions), neurophysiological assessment tools (ERP, qEEG, LORETA), cognitive outcome measures, and methodological aspects. In addition, although not pre-specified in the initial charting form, all studies were systematically reviewed for the use of modern spectral parameterization methods (e.g., FOOOF [23]) and for transparency and reproducibility practices (e.g., pre-registration, data sharing, code availability, detailed reporting of analysis pipelines; cf. [21,22]). These exploratory assessments were included to provide further insight into the analytical and methodological rigor of NFB research in sport.

The evaluation also focused on methodological strength through an assessment of sham controls (none, Active, or Inert), blinding procedures, and statistical approaches. Following [19,33,34], sham controls were operationally categorized as Active Sham—non-contingent but plausible feedback (e.g., pre-recorded EEG or randomized signals)—and Inert Sham, fully decoupled from participants’ physiological activity (e.g., random tones or pre-recorded videos). This classification allows for a clearer evaluation of methodological rigor, reducing the risk of conflating non-specific engagement effects with genuine NFB-related changes. A summary of the distribution of sham control types across studies is presented in the Section 3 (Figure 4) to providing a visual overview of this critical methodological factor.

When any essential information was unclear or unavailable, the corresponding study authors were contacted by email. However, response rates were limited, and missing data were coded as “not reported”.

Finally, the entire process was piloted on five studies to ensure consistency and clarity in the data extraction procedure, with particular emphasis on identifying the presence and type of sham controls as a critical methodological factor.

### 2.6. Synthesis of Results

The research findings will be summarized in tables that organize data by study design, population type (elite athletes, amateur athletes, non-athletes), and characteristics of the NFB protocols, as well as neurophysiological and cognitive outcomes.

A narrative synthesis will be conducted to highlight methodological trends, outcome patterns, and evidence gaps across studies. The synthesis will remain descriptive in nature, consistent with the scoping review methodology.

The analysis will also quantify the frequency of key methodological variables, including the presence and type of sham controls, blinding procedures, and EEG analysis techniques (e.g., qEEG, ERP, LORETA). These distributions will be reported to provide a structured overview of methodological rigor and transparency across the included studies.

## 3. Results

The scoping review analyzed 48 studies that examined EEG-based NFB interventions among athletes competing in various sports and at different competitive levels. The PRISMA 2020 flow diagram (Figure 1) illustrates the study selection process from identification through screening to final inclusion.

The included studies investigated athletes from a wide range of sports (e.g., archery, golf, gymnastics, swimming, soccer, judo, and chess) across multiple countries and competitive levels (elite, amateur, and novice).

The study characteristics are summarized in Table 1, which provides detailed information on authors, publication years, sample sizes, demographic characteristics, sport disciplines, study designs, electrode placement protocols, control group types, outcome measures, and reported intervention effects.

The extensive information presented in Table 1 serves as the primary reference for understanding the diversity and methodological scope of the included studies.

### 3.1. Selection of Sources of Evidence

The initial database search retrieved 3516 records, supplemented by an additional 240 records from gray literature and other sources. After removing 1737 duplicates, 1779 records remained for title and abstract screening. Of these, 1729 records were excluded based on eligibility criteria.

The full-text evaluation was conducted for 70 articles, of which 48 studies met the inclusion criteria and were included in the review. This corresponds to approximately 2.6% of the initially retrieved records.

The detailed selection process is presented in the PRISMA 2020 flow diagram (Figure 1). Reasons for full-text exclusion are documented in Appendix A, covering the 22 excluded studies.

Additionally, the distribution of electrode sites and frequency bands across the included studies is summarized in Figure 2.

Reason for exclusion at full-text stage:

Reason 1. Studies involving clinical populations (e.g., neurological or psychiatric diagnoses).

Reason 2. Studies lacking methodological detail on the neurofeedback protocol or outcomes.

Reason 3. Studies relying exclusively on subjective outcomes (e.g., self-perceived performance).

Reason 4. Non-peer-reviewed publications (e.g., conference abstracts, opinion articles, technical reports).

Reason 5. Full text not available or no response from corresponding authors after three contact attempts (within a three-week period).

The study selection process followed the PRISMA 2020 guidelines [67], as illustrated in Figure 1.

### 3.2. Characteristics of Included Studies

The included studies were conducted across 18 countries, with Poland contributing the largest share (24%), followed by Iran (18%) and Taiwan (12%). Other countries, including Germany, Portugal, and Canada, provided smaller but noteworthy contributions (Figure 3A,B).

In terms of research design, randomized controlled trials (RCTs) accounted for 60% of the studies, followed by quasi-experimental designs (29%) and case or single-subject approaches (11%) (Figure 3C). Most studies recruited participants ranging from novice to elite athletes, with males representing 77% of the total sample.

Control group strategies showed considerable variability: Active Sham conditions were used in 29% of studies, passive controls with no intervention in 33%, and no-control designs (e.g., pre–post or single-subject studies) in 38% (Figure 4 and Figure 5).

As shown in Figure 6, SMR-based training (12–15 Hz) was the most frequently applied protocol, followed by theta/beta and alpha-based modulation. This pattern underscores the predominance of SMR approaches in sports-related EEG-NFB research, reflecting their established association with motor control and attentional regulation. At the same time, the relatively lower prevalence of infra-low frequency, mu, and customized alpha- or ERP-based protocols highlights emerging directions that remain underrepresented in the current literature.

### 3.3. Neurophysiological Outcomes

Neurophysiological outcomes were reported in 52% (*n* = 25) of the included studies. The reported effects encompassed EEG spectral power changes, such as sensorimotor rhythm (SMR) enhancement at Cz (located at the vertex of the scalp, approximately over the sensorimotor cortex), and at C3 and C4 (positioned over the left and right primary cortices, respectively). Other studies examined ERPs, particularly components such as P3 and N2 [48], as well as coherence and connectivity measures derived from source localization techniques, including LORETA and sLORETA [37].

Studies that combined neurophysiological measures with behavioral assessments frequently reported associations between EEG changes and improvements in motor or cognitive performance [36,39]. The distribution of studies focusing on neurophysiological outcomes, compared with those relying exclusively on behavioral or cognitive assessments, is illustrated in Figure 7.

### 3.4. Cognitive Outcomes

Cognitive outcomes were reported in 89% (*n* = 43) of the included studies. The research primarily targeted three cognitive domains: attention, working memory, and executive functions. These were assessed through standardized paradigms such as inhibition tasks (e.g., Stroop test), working memory updating (e.g., N-back task), and cognitive flexibility/set-shifting (e.g., Oddball paradigm).

Standardized neuropsychological assessments—particularly the N-back task, Stroop test, and Oddball paradigm—were frequently complemented with sport-specific tasks, including reaction time tests in archery or golf putting performance, to evaluate cognitive improvements in real-world contexts.

Overall, the evidence indicated consistent cognitive benefits of neurofeedback training. For instance, studies highlighted improvements in attentional control [42,51], while others reported enhanced stress regulation and self-perceived mental readiness [35,57].

### 3.5. Methodological Features

The increasing focus on methodological rigor is reflected in the gradual adoption of randomized controlled trials (RCTs). Nevertheless, only 29% of studies (*n* = 14) included Active Sham feedback as a placebo control [38,64]. The majority of these Active Sham protocols (*n* = 15) relied on pre-recorded EEG data or randomized signals, which may still introduce unspecific neuroplastic changes [18,19]. Only three studies applied Inert Sham protocols that fully separated neural activity from feedback [44,45,46], representing the methodological gold standard for identifying neurofeedback-specific effects. Overall, approximately 40% of studies did not implement any sham control, relying on passive or no-control designs.

Beyond sham design, most studies did not report participant or evaluator blinding, and long-term follow-up assessments were rare (exceptions include [39,59]). These limitations further underscore the need for methodological consistency and transparency in EEG-NFB research.

Another critical issue concerns EEG spectral analysis. None of the 48 included studies applied modern spectral parameterization techniques such as FOOOF; [23]. Instead, all relied on conventional band-power approaches based on fixed frequency bands (e.g., SMR, alpha, theta, beta), typically calculated via Fast Fourier Transform (FFT). Some studies applied visual or manual inspection of EEG signals, and reporting of spectral analysis parameters was often incomplete. This reliance on traditional band-power metrics prevents separation of periodic oscillatory activity from the aperiodic 1/f background, which may bias the interpretation of NFB effects.

This methodological heterogeneity underscores the challenges of synthesizing evidence across studies and highlights the importance of adopting standardized protocols and transparent reporting practices. Figure 5, Figure 6 and Figure 7 illustrate these methodological inconsistencies, emphasizing the lack of sham standardization, limited neurophysiological outcome reporting, and the predominance of outdated spectral approaches.

### 3.6. Transparency and Reproducibility

The analysis of the 48 included studies revealed a systemic absence of open science practices. None of the studies provided data sharing, code availability, or preregistration. A single exception was noted in [62], which reported protocol approval by a local ethics committee prior to data collection; however, this does not constitute preregistration in the open science sense, as it lacked public accessibility and methodological detail.

Although most studies described their training protocols (e.g., electrode sites, frequency bands, session structures), independent replication remained unfeasible due to reliance on proprietary hardware/software and closed-source algorithms. In addition, statistical transparency was limited: the majority of studies reported only *p*-values, with rare mentions of effect sizes or confidence intervals, thereby constraining interpretability.

With respect to EEG analysis, all studies relied on traditional band-power metrics in fixed frequency bands. None applied modern spectral parameterization methods such as FOOOF [23], which separate periodic oscillatory activity from the aperiodic 1/f background. This reliance on fixed-band approaches—often embedded in commercial systems—further restricts the neurophysiological validity of reported outcomes.

Taken together, these findings align with concerns raised by [21,22], highlighting the urgent need for open data, shared code, preregistration, and transparent reporting of analytic pipelines to ensure reproducibility and credibility in EEG-NFB research.

## 4. Discussion

This scoping review synthesized 48 studies examining EEG-based NFB interventions across athletic and non-athletic populations. The evidence generally supports the potential of NFB to modulate neurophysiological activity and improve cognitive and performance-related outcomes. However, the review also exposes substantial methodological heterogeneity and reproducibility gaps that complicate interpretation and cross-study comparison. The following sections discuss these findings considering previous literature, highlighting consistent trends, discrepancies, and future research needs.

### 4.1. Neurophysiological and Cognitive Outcomes

Across the analyzed studies, NFB training most frequently targeted SMR and alpha bands, with reported increases in EEG power often corresponding to improvements in reaction time, attention, and motor precision. These findings align with early work by [2,3], who demonstrated that modulating SMR and alpha activity could facilitate motor preparation and cognitive stability. Similarly, more recent studies—such as [36,37]—confirmed enhanced motor accuracy and balance following SMR- and theta/beta-based training, supporting the link between neural regulation and performance optimization.

Nevertheless, not all evidence converges. Some experiments, such as [43], reported null effects on reaction time or inconsistent EEG modulation, suggesting that task specificity, participant expertise, and feedback parameters critically influence outcomes. Cognitive measures—particularly attention, working memory, and executive control—were the most frequently improved domains, in line with systematic syntheses by [4,5]. Yet, the diversity of testing paradigms (e.g., Stroop, N-back, Oddball) and the predominance of short-term assessments limit the generalization of these results. Overall, the current evidence indicates that EEG-NFB can induce measurable neural and behavioral adaptations, though magnitude and persistence remain uncertain due to methodological inconsistency.

### 4.2. The Role of Sham Controls

A central concern identified in this review involves the design and implementation of sham controls. As defined in Section 2.5, Active Sham refers to non-contingent but plausible feedback, whereas Inert Sham is fully decoupled from participants’ physiological activity. Only 29% of studies employed active sham feedback, and a mere 6% applied inert sham protocols—the methodological gold standard for isolating true NFB-specific effects. This distinction aligns with the CRED-nf recommendations [20] and prior methodological reviews [33], which emphasize the need for transparent reporting of sham procedures in EEG-NFB research. These proportions mirror the shortcomings previously highlighted by [19], who emphasized that expectancy and engagement effects may inflate apparent efficacy in NFB research. The scarcity of inert sham conditions observed here suggests that many studies risk conflating neurophysiological change with non-specific psychological factors.

Furthermore, a large subset of studies lacked participant or assessor blinding and relied solely on pre–post comparisons. Such designs increase susceptibility to placebo effects and Type III statistical errors, as discussed by [34]. When properly implemented, double-blind randomized trials—such as those by [35] or [38]—demonstrated more controlled evidence for EEG modulation and performance enhancement. Future investigations should therefore integrate both active and inert sham conditions, coupled with rigorous blinding, to strengthen internal validity and permit clearer attribution of causal effects.

### 4.3. Electrode and Frequency Variability in Neurofeedback Protocols

The diversity of electrode montages and targeted frequency bands across the reviewed studies reflects the absence of standardized NFB protocols in sport settings. Central sites (Cz, C3, C4) were predominant in SMR-based interventions, consistent with their functional relevance to motor preparation and attention. However, frontal and parietal placements targeting alpha, beta, or theta activity were also frequent, often motivated by exploratory aims rather than established neurophysiological models. Comparable variability was reported by [2,4], who noted that inconsistency in training loci and spectral ranges impedes replication and cumulative synthesis.

This heterogeneity complicates the interpretation of EEG changes and performance outcomes. Even when similar behavioral gains were reported, underlying neural mechanisms may differ due to protocol divergence. Standardized reporting through frameworks such as the CRED-nf checklist [20] and adoption of modern spectral parameterization tools like FOOOF [23] would allow more accurate separation of oscillatory and aperiodic components, thereby improving cross-study comparability and theoretical precision.

### 4.4. Ecological Validity and Implementation Challenges

Recent studies increasingly integrate portable EEG systems and field-based protocols to enhance ecological validity and bridge laboratory findings with real-world athletic contexts. Investigations by [25,26] illustrate this trend, demonstrating that brief, on-site SMR training sessions can positively influence golf and soccer performance. These developments parallel broader efforts in applied neuroscience to situate cognitive training within authentic performance environments.

However, ecological validity introduces methodological complexity. Field-based EEG is inherently vulnerable to motion artifacts, environmental noise, and fatigue effects that can compromise data quality and mask the specific contribution of NFB. Moreover, most reviewed studies relied on short-term pre–post designs without longitudinal follow-up, precluding conclusions about retention or transfer of NFB benefits. Sustained improvements, as observed in long-term follow-ups by [59], remain rare but essential for verifying whether NFB-induced adaptations persist beyond initial training phases. Future research should therefore combine controlled laboratory paradigms with extended, ecologically grounded interventions, using multimodal outcome measures (EEG, qEEG, ERP, behavioral, and psychophysiological indices) to achieve comprehensive evaluation.

### 4.5. Considerations for Future Research

To consolidate the evidence base for EEG-NFB in sport, future investigations must emphasize methodological rigor, analytical transparency, and ecological realism. Randomized controlled trials incorporating both active and inert sham conditions are imperative to distinguish genuine neurofeedback effects from non-specific influences. Protocol standardization regarding electrode placement, targeted frequency bands, and session parameters will facilitate replication and meta-analytic synthesis.

Equally crucial is the transition toward open-science practices. None of the reviewed studies preregistered protocols or shared data and analysis code, reflecting a broader reproducibility gap in applied neuroscience [21,22]. Adopting preregistration, data sharing, and transparent reporting of analytic pipelines will markedly enhance credibility and cumulative progress. Aligned with the CRED-nf checklist [20], future EEG-NFB studies should explicitly preregister core components of their experimental design, including hypotheses, primary and secondary outcomes, session parameters, and planned statistical analyses. Minimal datasets—such as pre-processed EEG spectra, behavioral measures, and analytic scripts—should be made openly available in public repositories (e.g., OSF, Zenodo, OpenNeuro). Moreover, the transparent reporting of key signal-processing parameters (e.g., FFT settings, filter characteristics, artifact rejection thresholds, and reinforcement schedules) will facilitate methodological reproducibility and cross-study comparability. Collectively, these practices will transform general calls for transparency into concrete, actionable standards for advancing open science in EEG-NFB research.

Finally, research should expand participant diversity—addressing gender balance and sport variety—and include longitudinal follow-ups to determine the durability and ecological transfer of NFB-induced performance gains. By integrating these methodological and conceptual refinements, future studies can transform EEG-NFB from a promising experimental approach into a reproducible, evidence-based tool for optimizing human performance.

Another potential methodological concern involves the partial overlap of samples across studies conducted by the same research groups (e.g., [46,48,57,60,61]). Such overlap may inflate the apparent evidence base and reduce the effective sample diversity, particularly when similar participant cohorts are repeatedly analyzed under slightly modified protocols. This limitation should be considered when interpreting the overall findings, as it may bias outcome generalizability and overestimate the robustness of specific training effects. Future reviews should therefore apply stricter data-source screening procedures and explicitly report instances of potential sample duplication to enhance the transparency and reproducibility of evidence synthesis in EEG-NFB research.

## 5. Conclusions

EEG-based NFB demonstrates meaningful potential to enhance both neurophysiological regulation and cognitive-motor performance in athletes. Yet, this promise remains constrained by inconsistent methodology, limited sham control, and insufficient transparency. The field now requires rigorously designed, double-blind randomized trials using validated sham procedures and standardized spectral analyses to establish causal validity.

Future progress depends equally on adopting open-science principles—preregistration, data and code sharing, and clear protocol reporting—to ensure replicability and comparability across studies. Long-term, ecologically valid designs will clarify whether short-term NFB effects translate into sustainable performance benefits. Strengthening methodological rigor and transparency will not only improve scientific reproducibility but also enable NFB to fulfill its potential as a practical tool in sport neuroscience.

## Figures and Tables

**Figure 1 bioengineering-12-01202-f001:**
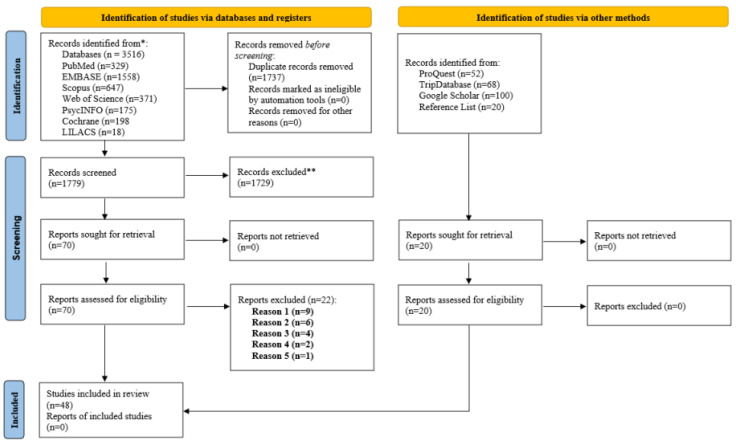
PRISMA-ScR flow diagram showing the identification, screening, and inclusion of the studies included in this scoping review. * Records identified from the listed electronic databases. ** Records excluded after title and abstract screening.

**Figure 2 bioengineering-12-01202-f002:**
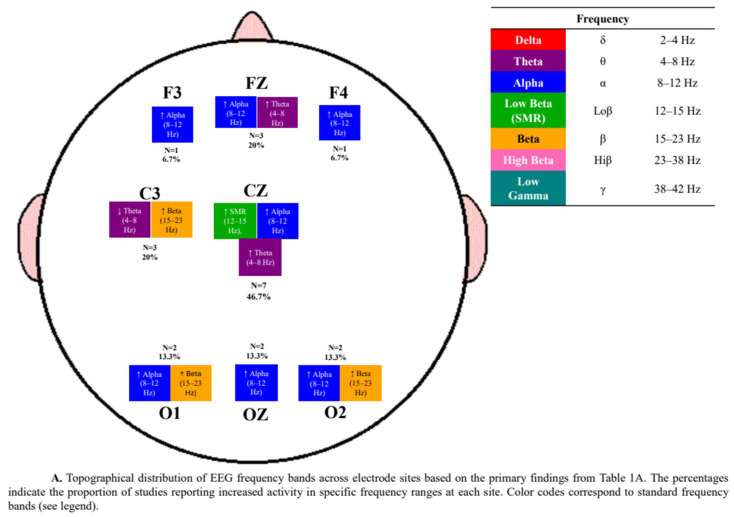
Distribution of electrode sites and frequency bands in studies using different neurofeedback protocols: (**A**) Active Sham neurofeedback (Table 1A); (**B**) Inert Sham neurofeedback (Table 1B); and (**C**) without Sham neurofeedback (Table 1C).

**Figure 3 bioengineering-12-01202-f003:**
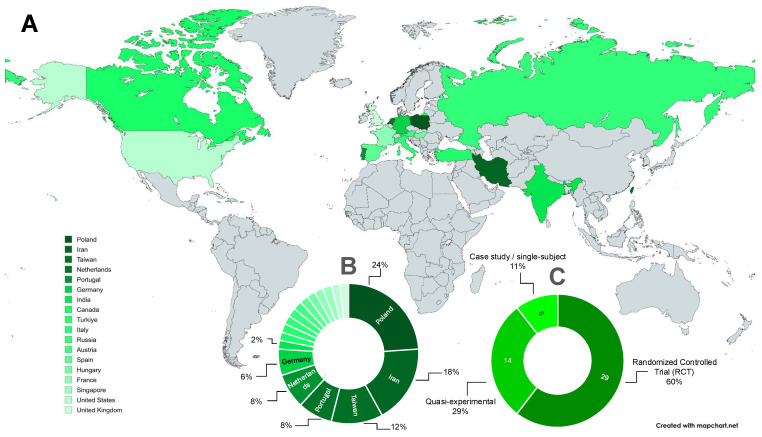
Characteristics of included studies: (**A**) Geographic distribution; (**B**) Country contributions by percentage; (**C**) Study design classification.

**Figure 4 bioengineering-12-01202-f004:**
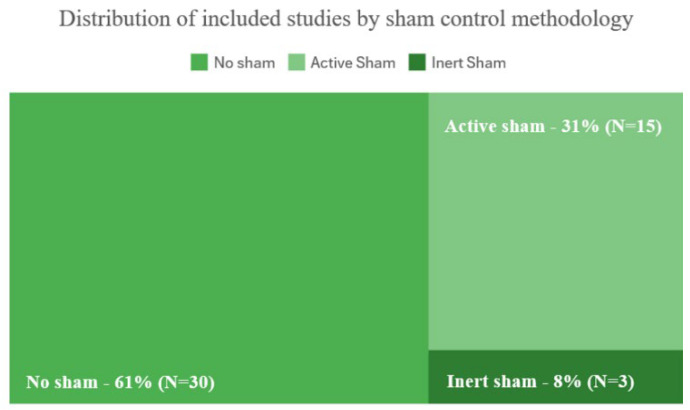
Proportion of studies using different sham control types (no sham, active sham, and inert sham).

**Figure 5 bioengineering-12-01202-f005:**
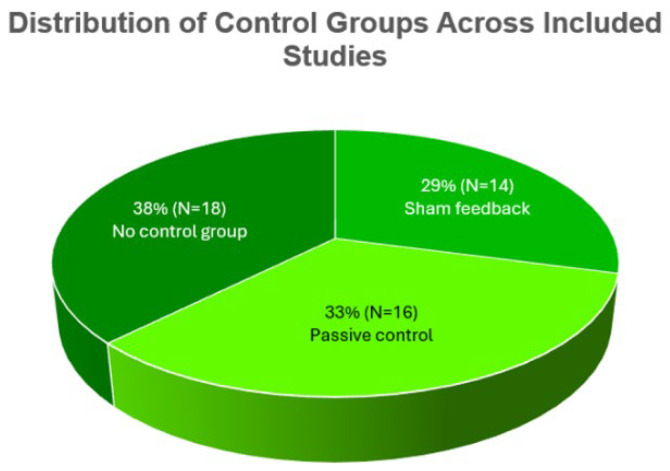
Distribution of control group types across included studies (sham feedback, passive control, no control).

**Figure 6 bioengineering-12-01202-f006:**
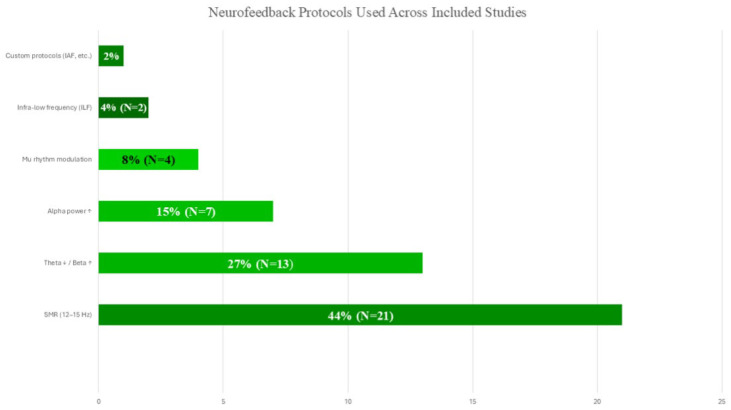
Neurofeedback protocols used across included studies (SMR, theta/beta, alpha, ILF, um rhythm, custom).

**Figure 7 bioengineering-12-01202-f007:**
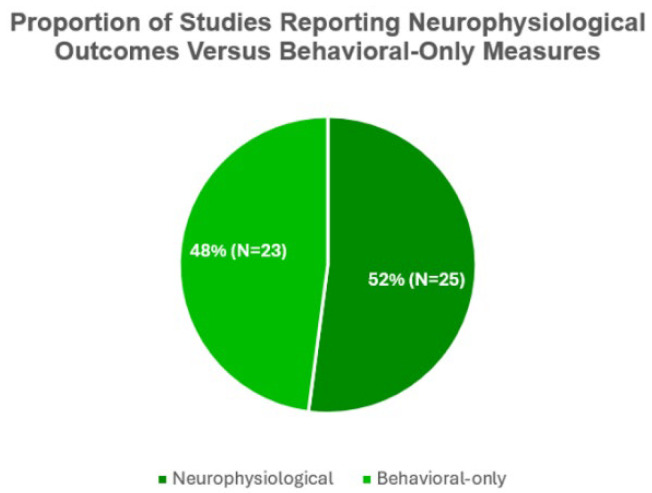
Proportion of studies reporting neurophysiological outcomes versus behavioral-only measures.

**Table 1 bioengineering-12-01202-t001:** (**A**) Studies with Active Sham. Presents studies with *Active Sham* (feedback from pre-recorded EEG or randomized signals). (**B**) Studies with Inert Sham. Shows studies that used feedback completely unrelated to EEG (Inert Sham). (**C**) Studies without Sham. Includes studies with no form of sham (only passive control groups or pre-post designs).

**(A)**				
**Authors** **(Year)**	**Sample**	**Discipline & Level**	**Protocol (Summary)**	**Main Findings**
[6]	24 pre-elite archers	Archery—pre-elite with competition experience	RCT with 3 groups: correct feedback, incorrect (active sham), and control; feedback modeled on slow cortical potential (SCP) paradigm	Significant improvement in performance in performance in the correct feedback group; performance decrement in incorrect feedback group; no significant change in control
[35]	12 elite gymnasts	Gymnastics—national/international	Double-blind RCT: Alpha-band NFB vs. Sham (active control with random beta-band feedback)	Trend toward improved mental balance, physical shape, and reduced sleep complaints (ns)
[36]	16 pre-elite/elite golfers	Golf—national/international	RCT: SMR-based NFB vs. Sham; pre-post assessment of EEG and performance	Increased putting accuracy, consistency, and SMR power in NFB group; no change in control
[37]	18 elite judokas	Judo—national/international	Double-blind RCT: NFB targeting θ/β ratio vs. Sham; pre-post design	Significant gains in balance and β-power (*p* < 0.05); no significant change in sham group
[7]	30 male basketball players	Basketball—competitive athletes (level not specified)	Double-blind quasi-experimental design: generic NFB protocol (EEG at Cz/Fz/parietal); control with sham feedback	Improved reaction time, balance, attention and reduced anxiety compared with controls (*p* < 0.05)
[38]	26 triathletes (elite endurance) and 25 controls	Triathlon—elite endurance athletes	Double-blind RCT: SMR-NFB vs. Sham (single session); EEG and MRI metrics	Real NFB group: ↑ SMR power (*p* = 0.02), ↑ gray/white matter volumes (*p* < 0.001); no change in sham
[39]	64 novice golfers	Golf—no prior competitive experience	RCT: SMR, alpha, mu NFB vs. Sham; pre-post and retention tests	SMR & alpha groups: ↑ putting accuracy and EEG power short- and long-term; mu group ≈ sham
[40]	36 skilled golfers	Golf—competitive amateurs	Single-session RCT: FSI-NFB vs. TI vs. Sham; FMT measured	FSI-NFB improved putting success and reduced FMT power (*p* < 0.05); sham and TI showed no improvement
[41]	40 recreational cyclists	Cycling—recreational	Crossover RCT: NFL-NFB vs. NFR-NFB vs. Sham; endurance and EEG metrics assessed	NFB increased endurance (+30%, *p* < 0.05), frontal alpha asymmetry, and HR/RPE; no changes in lactate or cadence
[42]	40 novice golfers	Golf—no prior experience	RCT: SMR-NFB vs. Sham; with self-controlled and yoked conditions; pre-post follow-up design	SMR-NFB group: ↑ putting accuracy (*p* < 0.05), ↑ SMR power, ↑ self-control; no additive effect beyond feedback
[43]	38 male soccer players (14–23 y; ≥4 y training)	Soccer—competitive athletes	RCT: TBR-NFB and SMR-NFB vs. Sham; 10 sessions; outcomes: attention, reaction time, EEG power	No significant improvement in attention or reaction time; no EEG changes reported
[8]	12 elite judo athletes (Polish Judo Assoc.)	Judo—elite athletes	RCT: β↑/θ↓ NFB vs. Sham; 15 sessions across 2 cycles	NFB group showed ↑ reaction speed and β power, ↓ θ power; sham group showed no change
[24]	24 recreational golfers (all male)	Golf—novice athletes	RCT: high-alpha/theta NFB vs. Sham; 3 sessions; putting accuracy task	NFB group: ↓ frontal high-alpha power; no selective performance improvement; both groups performed similarly under pressure
[9]	12 elite judokas	Judo—elite national	RCT: β↑/θ↓ NFB vs. Sham; 15 sessions in 2 training cycles	NFB group: ↑ reaction speed and β power, ↓ θ power; sham group unchanged
[10]	12 elite judokas	Judo—elite international	RCT: β↑/θ↓ NFB vs. Sham; 15 sessions (2 cycles); system: Deymed TruScan (Deymed Diagnostic-Hronov, Czech Republic)	NFB group: ↑ reaction speed and β power, ↓ θ power; no EEG change in sham
**(B)**				
**Authors** **(Year)**	**Sample**	**Discipline & Level**	**Protocol (Summary)**	**Main Finding**
[44]	30 novice golfers (15 F, 15 M)	Golf—no prior experience	RCT: ↑ Mu-NFB vs. ↓ Mu-NFB vs. Sham; single session (BioTrace+ system. Mind Media, Herten, The Netherlands); pre–post design	↓ Mu-NFB group: ↓ Mu power, ↓ MRE, ↑ putting accuracy (*p* = 0.006); ↑Mu-NFB showed no significant changes; sham had no effect
[45]	31 young athletes (multi-sport)	Multi-sport—regular training	Single-blind RCT: SMR/Theta NFB vs. Sham; 12 sessions over 4 weeks (NeuroTracker platform, NeuroTrackerX Inc., Montreal, QC, Canada)	All groups improved cognitive performance (*p* < 0.05), but no between-group differences and no EEG changes
[46]	30 novice golfers (14 F, 16 M)	Golf—no prior experience	Double-blind RCT: FSI-NFB vs. TI-NFB vs. Sham; single session (BioTrace+)	FSI-NFB group: ↓ putting accuracy (*p* = 0.013), slight ↑ Mu power (ns), positive Mu–error correlation (r = 0.319, *p* = 0.043); TI and sham showed no changes
**(C)**				
**Authors** **(Year)**	**Sample**	**Discipline & Level**	**Study Design**	**Electrode & Protocol**
[47]	6 amateur golfers (3 F, 3 M; avg handicap 12.3)	Golf—amateur	Quasi-experimental within-subject ABAB design; same athletes performed alternating blocks with and without event-locked NFB (Fpz channel)	NFB blocks led to ↑ putting success (+25%) compared to no-NFB blocks; no sham group included
[48]	1 elite javelin thrower (25 y, Olympic level)	Javelin—elite	Case study using alpha-NFB (C3/C4; ProComp Infiniti, Thought Technology Ltd., Montreal, QC, Canada); ERP assessment (NOGO task)	Improved attention and social behavior; ↓ aggression; ↑ ERP changes and reaction time
[49]	1 elite javelin thrower (male, 25 years old, Olympic-level athlete)	Javelin—elite Olympic athlete (2012 London Olympics participant)	Single-subject pre–post case study; Alpha/Low-Beta NFB (C3/C4; ProComp Infiniti); ERP and cognitive testing	↑ ERP amplitudes and β power; improved cognitive control and emotional regulation
[50]	3 expert golfers (20–25 y, >10 y exp.)	Golf—elite	Case study with multiple baseline design (3 athletes); EEG at Fz (4–8 Hz) downregulation; NeuroTek training (NeuroTek, Goshen, KY, USA)	2/3 participants: ↑ putting performance and ↓ θ power; mixed results for anxiety and confidence
[51]	35 semi-professional athletes (25 M, 10 F; mixed sports)	Multi-sport—semi-pro	Quasi-experimental: 20 NFB sessions (alpha, beta1, theta) vs. control; pre–post design	EG showed ↑ α and β1 power, faster reaction time, improved mental performance; control group unchanged; no sham group
[52]	73 student athletes (40 M, 33 F; 18–25 y; swimming, fencing, track & field, taekwondo, judo)	Multi-sport—national-level	Quasi-experimental: 20 NFB sessions (SMR, beta1, theta) vs. control; 7-month program	EG: ↑ SMR & β1, ↓ θ, ↑ mental readiness/performance; CG: no change
[11]	21 elite athletes (soccer & track & field; 16–38 y)	Soccer & Track & Field—elite	Quasi-experimental: alpha↑ NFB (C3/C4); 4 coached + daily home sessions (5 weeks)	Soccer group: ↑ alpha (5/7 EEG sites), ↓ LF/HF, ↑ emotional stability, concentration & sleep; Track & Field: ↑ recovery (RESTQ), sustained LF/HF balance
[12]	5 elite female rifle shooters	Rifle Shooting—elite international	Single-subject design using β1/θ NFB at P8 (Emotiv 14-channel EEG System, San Francisco, CA, USA); 6–7 sessions; attention and shooting accuracy assessed	3/5 athletes improved shooting accuracy and attention; 2 remained stable; no adverse effects reported
[53]	18 elite handball players (9 M, 9 F; 1st/2nd league)	Handball—elite	Quasi-experimental NFB (↑SMR, ↑β1, ↓θ, ↓β2) using DigiTrack system (ELMIKO Medical, Warsaw, Poland); 20 sessions over 10 weeks; outcomes: attention, sensorimotor coordination, peripheral perception	Improved attention, ↑ sensorimotor coordination, ↑ peripheral perception (mainly in males)
[54]	45 participants (15 athletes, 15 non-athletes, 15 controls; 18–44 y)	Multi-sport—athletes & non-athletes	RCT: athletes-NFB, athletes-control, non-athletes-NFB, non-athletes-control; 12–15 sessions; EEG at Cz (Vertex 823 system—Meditron Eletrônica, São Paulo, Brazil)	Athletes-NFB group: ↑ reaction time and IAB; non-athletes-NFB: ↑ SAB/IAB; control groups: no change
[13]	7 elite swimmers (~20.6 y)	Swimming—elite	Pre–post single-group design (no control); NFB at C3/C4 (TruScan Flex 30); β (20–30 Hz) inhibition and SMR enhancement; 20 sessions (6 × 5 min) during exercise across 4 months	↑ mental work capacity, ↑ EMG signal consistency; no changes in VO_2_max or anaerobic performance
[55]	45 student athletes (7 F, ages 18–35; ≥5 y practice)	Multi-sport—student athletes	RCT: noisy vs. silent-room NFB vs. control; 12 sessions; working memory and reaction time tasks	Noisy-NFB group: ↑ working memory (*p* = 0.005) and faster reaction time; silent NFB and control: no effect
[56]	45 male student athletes (18–34 y; ≥5 y practice)	Multi-sport—student athletes	RCT: 3×/week vs. 2×/week NFB vs. control; 12 sessions; EEG at Cz (Meditron Vertex 823)	3×/week group: ↑ IAB and faster reaction time; 2×/week and control: no significant changes
[57]	30 male student athletes (18–34 y; ≥5 y practice)	Multi-sport—student athletes	Quasi-experimental: 3×/week vs. 2×/week NFB; 12 sessions; HRV and IAB measured	3×/week group: ↑ IAB and HRV; 2×/week: no change
[14]	15 male professional soccer players (17.6 y; U17/U19/N2 levels)	Soccer—pro youth	Single-group RCT (no sham); EEG at P3/P4; 7 × 3-min sessions (Spectre Biotech, Suresnes, France)	↑ attention and reaction (+30%, +27%, *p* < 0.01); effect maintained at 1-month follow-up (+20%)
[58]	20 athletes (10 track & field, 10 swimmers; 18–25 y)	Multi-sport—elite athletes	Quasi-experimental: β2-NFB at C3/C4; 20 sessions over 4 months; EEG and recovery measures	↑ EEG (θ–β) modulation during attention tasks, ↑ effort and recovery; control group: no change
[59]	1 elite female chess player (ELO > 2350; Top 100 worldwide)	Chess—elite athlete	Case study: SMR↑/θ↓ NFB + BFB; 14 sessions + 6/12-month follow-up	↑ chess performance (+38% puzzle rush, +20–25 ELO), ↓ anxiety, ↑ HRV and regulation control; sustained gains at 6–12 months
[25]	41 elite soccer players (26 F, 15 M; 2 Dutch pro teams; ~20 s)	Soccer—elite national	Crossover quasi-experimental study: α↑ NFB (BrainBit music-feedback system, BrainBit Inc., Rancho Santa Margarita, CA, USA); 20 sessions over 4–6 weeks; cognitive tasks included (N-back, PVT, mental rotation)	EG: ↑ α power (+34%, *p* < 0.001), ↑ task switching and mental rotation performance; slight ↑ in control/flow; no effect on N-back or PVT
[26]	44 professional golfers (20 F, 24 M; PGA/LPGA; mean age 26.8 y)	Golf—professional	Crossover RCT: SMR-NFB vs. no-training control; single session (~2.5 h); EEG at Cz (12–15 Hz; ProComp5 Infiniti)	NFB group: ↑ putting accuracy and SMR power (*p* < 0.01), ↑ attention/motor control, ↑ relaxation (*p* < 0.01); control: no change
[27]	17 professional female golfers (PGA of Taiwan/LPGA; mean age ~24.6 y)	Golf—professional	Crossover RCT: SMR-NFB vs. no-training control; single session (~2.5 h) using ProComp5 Infiniti; EEG at Cz	NFB group: ↑ swing accuracy (To Pin, *p* = 0.04), ↑ SMR power, ↑ motor control and relaxation; control group: no change
[60]	20 semi-skilled pistol shooters (10 M, 10 F; mean age 28–40 y)	Shooting—semi-skilled national level	RCT: ILF-NFB vs. control; 20 sessions over 7 weeks; EEG at T3/T4/P4/Fp1	ILF-NFB group: ↑ shooting accuracy (*p* = 0.005), ↑ attention network efficiency (ANT, *p* < 0.01); control: no change
[61]	24 university archers (16 M, 8 F; ~22 y; 4 y experience)	Archery—university competitive	RCT: SMR-NFB vs. control; 12 sessions over 4 weeks (ProComp5 Infiniti); EEG at Cz	SMR-NFB group: ↑ pleasure and arousal (*p* < 0.05), ↓ SMR/θ ratio (*p* < 0.05); no performance improvement; control group: ↓ precision over time
[62]	24 expert rifle shooters (~30 y; ~7 y experience)	Shooting—expert, national/provincial	Pre–post design: NFB vs. control; 15 sessions over 5 weeks (ProComp P2 and P8 systems (Thought Technology Ltd., Montreal, QC, Canada; C3/Pz; SMR, β1↑, α↑)	NFB group: ↑ shot result (*p* = 0.001); no other EEG or behavioral changes; control: no improvement
[63]	28 female gymnasts (15 NFB, 13 control)	Gymnastics—high-skilled, competitive period	Pre–post quasi-experimental design: α↑ NFB (F1/F2/P3/P4; Boslab-alpha (SRIMBB RAMS, Novosibirsk, Russia)); 15 sessions before competition	NFB group: ↑ α power (*p* < 0.05), ↑ vestibular stability, ↑ memorization speed, ↓ self-estimation bias; control: no change in attention/anxiety
[15]	20 national-level pistol shooters (3 F, 17 M)	Pistol Shooting—national-level experts	RCT: α↑ NFB vs. control; 16 sessions (Peak Achievement Trainer)	NFB group: ↑ shooting accuracy (*p* < 0.001), ↑ temporal α power; no change in coherence; control: no improvement
[64]	31 university ice hockey players (18 F, 13 M; ~21.7 y)	Ice Hockey—elite university athletes	RCT: SMR/θ/β1 NFB + BFB vs. control; 15 sessions over 4.5 months	NFB group: ↑ shooting accuracy (*p* = 0.018), ↑ SMR (*p* = 0.001); no SMR change during competition; control: slower improvement
[65]	45 novice basketball players (3 groups, *n* = 15)	Basketball—novice athletes (1–3 y exp.)	RCT: BFB + NFB vs. BFB vs. control; 24 sessions over 8 weeks (ProComp Infiniti system)	BFB + NFB group: ↑ technical skills (lay-up, passing) and physiological indices (BFB, HRV); control: no improvement
[66]	20 elite female swimmers (13–14 y; 5–6 y experience)	Swimming—elite	RCT: SMR↑, θ↓, high-β↓ NFB (videogame-based) vs. control; 12 sessions over 4 weeks	NFB group: ↓ anxiety (*p* < 0.01); control: no change; significant time × group interaction (*p* = 0.017)
[16]	30 male volleyball players (15 elite, 15 non-elite; mean age 22.8 y)	Volleyball—elite & non-elite	Quasi-experimental: 1 NFB session (SMR↑ 12–15 Hz; T3, T4 sites; videogame feedback); comparison between elite and non-elite groups	↓ self-talk (*p* < 0.05), ↑ serve performance (*p* < 0.01); greater gains observed in elite athletes
[17]	24 female rugby players (16–25 y; *n* = 12 NFB, *n* = 12 control)	Rugby—female athletes	Quasi-experimental pre–post design: NFB vs. control; 15 sessions (3×/week, 40 min); α↑ at Pz and SMR↑ at C3	NFB group: ↑ passing accuracy (*p* < 0.01, both sides); no change in shot accuracy; control: no improvement

**Abbreviations and symbols:** ABAB design—alternating no-feedback/feedback block sequence; α—alpha band; β—beta band; ANT—Attention Network Test; BFB—biofeedback; C3, C4, Cz, Fpz, Fz, P4, P8, Fp1—standard EEG electrode sites (10–20 system); CG—control group; EG—experimental group; ELO—Elo rating system (FIDE chess ranking method); FMT—frontal midline theta; FSI-NFB—function-specific instruction neurofeedback; HR—heart rate; HRV—heart rate variability; IAB—individual alpha band; ILF—infra-low frequency; LF/HF—low-frequency to high-frequency power ratio; MRI—magnetic resonance imaging; MRE—mean radial error; NFB—neurofeedback; ns—not significant; PVT—Psychomotor Vigilance Task; RCT—randomized controlled trial; RESTQ—Recovery-Stress Questionnaire for Athletes; RPE—rating of perceived effort; SAB—standard alpha band; SMR—sensorimotor rhythm; θ—theta band; TI-NFB—traditional instruction neurofeedback; ↑—increase; ↓—decrease.

## Data Availability

All data supporting the findings of this study are included within the article and its Appendix A.

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
