# Peer review of "EEG-Based Neurofeedback in Athletes and Non-Athletes: A Scoping Review of Outcomes and Methodologies"

_bioengineering, 2025, doi:10.3390/bioengineering12111202_

Round 1
Reviewer 1 Report
Comments and Suggestions for Authors
Dear Authors,
First, I would like to commend you for your efforts in conducting this important study. The manuscript presents a clearly described scoping review on the methodological inconsistencies and reproducibility gaps of EEG neurofeedback applications. While the study provides valuable insights, several areas require clarification and revision before the manuscript can be considered for publication. Please find my detailed comments below.
Abstract
-
L11: Please define “EEG” at first mention.
-
L15: Add the main objective of the study before presenting the methods.
-
L16: Be more specific when describing “seven academic and gray literature sources.”
Introduction
-
The introduction would benefit from further development. In particular, please provide more detail on EEG neurofeedback applications in athletes, which is currently underrepresented.
-
Several paragraphs are very short, making the flow fragmented. Consider merging and expanding them to provide deeper context.
-
Improve transitions between paragraphs to enhance coherence.
-
L38: “Despite encouraging findings” – please specify which findings and provide citations.
-
L53: Add a supporting citation.
-
L54–56: The definition of “ecological validity” is too brief. Expand this section.
-
L57–60: Present the study objectives more clearly and explicitly.
Materials and Methods
-
The section is well structured and detailed.
-
L72–77: Please provide references for the classification of athletes, or cite a comprehensive source covering all categories.
-
Specify the keywords used in the article search strategy.
Results
-
The results are generally well presented. However, the tables include an excessive number of study examples, which makes them difficult to follow. Consider summarizing or grouping studies more concisely.
-
In Figure 1 (“Records identified from”), some underlined words appear to be part of a screenshot. Please revise for consistency and clarity.
Discussion
-
The discussion is clearly organized, with useful subtitles.
-
However, the results are not sufficiently compared with relevant literature. This is a critical weakness for a scoping review and reduces the overall quality of the paper. Please expand this aspect.
-
L380–396: Avoid itemization in the discussion section. Consider rewriting this part in a traditional narrative format.
-
Consider shortening the conclusions section to make it more focused and concise.Best wishes,
Author Response
Please see the detailed point-by-point responses provided in the attached document titled
“Response to Reviewer 1”.
All revisions corresponding to each comment are clearly highlighted in yellow in the revised manuscript.

Reviewer 2 Report
Comments and Suggestions for Authors
- The distinction between "Active Sham" and "Inert Sham" is crucial for the review's conclusions. It would be beneficial to provide a clearer rationale for these categories, perhaps with a sentence or two defining each type early in the results or methods section. For example, specify that "Inert Sham" involves feedback completely decoupled from the participant's physiology (e.g., random tones, pre-recorded videos), while "Active Sham" uses non-contingent but plausible neurofeedback (e.g., pre-recorded EEG from another participant). This will help readers better understand the methodological rigor being assessed.
- The figures (e.g., Figures 2A-C, 4, 5, 6, 7) are valuable but could be better integrated into the main text. The results section often presents percentages that are also visualized in the figures, leading to some redundancy. Consider streamlining the text by referring the reader directly to the figures for specific distributions and focusing the narrative on interpreting the key patterns and their implications. For instance, when stating that SMR training was the most frequent protocol (44%), directly refer to Figure 6 and discuss what this prevalence means for the field.
- The discussion briefly mentions the potential issue of sample overlap between studies from the same research groups (e.g., references [33,35,47,54,55]). This is an important methodological point that deserves a more detailed discussion. The authors should elaborate on how this potential overlap might inflate the evidence base or introduce bias, and consider discussing the implications for interpreting the overall findings of the review. Acknowledging this limitation more thoroughly will enhance the critical depth of the analysis.
- The section correctly identifies a near-total absence of open science practices. To make this critique more actionable, the authors could provide more specific, concrete recommendations aligned with existing guidelines like the CRED-nf checklist [7], which is mentioned but not fully leveraged. For example, suggest which specific aspects of a protocol should be pre-registered, what minimal data (e.g., pre-processed spectra) should be shared, and which analysis parameters (e.g., FFT settings, artifact rejection thresholds) are critical to report. This will transform a general criticism into a clear roadmap for future studies.
Author Response
Please see the detailed point-by-point responses provided in the attached document titled
“Response to Reviewer 2”.
All revisions corresponding to each comment are clearly highlighted in green in the revised manuscript.
